

# Mobile-UI-Repair: a deep learning based UI smell detection technique for mobile user interface

Asif Ali[1], Yuanqing Xia[1,2], Qamar Navid[1], Zohaib Ahmad Khan[1], Javed Ali Khan[3], Eman Abdullah Aldakheel[4] and Doaa Khafaga[4]

[1] School of Automation, Beijing Institute of Technology, Beijing, China
[2] Zhongyuan University of Technology, Zhengzhou, Henan, China
[3] Department of Computer Science, Faculty of Physics, Engineering, and Computer Science, University of Hertfordshire, Hatfield, United Kingdom
[4] Department of Computer Sciences, College of Computer and Information Sciences, Princess Nourah bint Abdulrahman University, Riyadh, Riyadh, Saudi Arabia

Corresponding author
Yuanqing Xia,
xia_yuanqing@bit.edu.cn

## ABSTRACT

The graphical user interface (GUI) in mobile applications plays a crucial role in connecting users with mobile applications. GUIs often receive many UI design smells, bugs, or feature enhancement requests. The design smells include text overlap, component occlusion, blur screens, null values, and missing images. It also provides for the behavior of mobile applications during their usage. Manual testing of mobile applications (app as short in the rest of the document) is essential to ensuring app quality, especially for identifying usability and accessibility that may be missed during automated testing. However, it is time-consuming and inefficient due to the need for testers to perform actions repeatedly and the possibility of missing some functionalities. Although several approaches have been proposed, they require significant performance improvement. In addition, the key challenges of these approaches are incorporating the design guidelines and rules necessary to follow during app development and combine the syntactical and semantic information available on the development forums. In this study, we proposed a UI bug identification and localization approach called Mobile-UI-Repair (M-UI-R). M-UI-R is capable of recognizing graphical user interfaces (GUIs) display issues and accurately identifying the specific location of the bug within the GUI. M-UI-R is trained and tested on the history data and also validated on real-time data. The evaluation shows that the average precision is 87.7% and the average recall is 86.5% achieved in the detection of UI display issues. M-UI-R also achieved an average precision of 71.5% and an average recall of 70.7% in the localization of UI design smell. Moreover, a survey involving eight developers demonstrates that the proposed approach provides valuable support for enhancing the user interface of mobile applications. This aids developers in their efforts to fix bugs.

# INTRODUCTION

User interface (UI) is crucial to modern software applications and mobile devices. It acts as a visual bridge between the application and its users, allowing them to interact with the software. A well-designed UI incorporates effective user interaction, clear information architecture, and engaging visual effects. Therefore, a good GUI design can greatly enhance an application's usability, efficiency, and overall success which also fosters user loyalty (*Haering, Stanik & Maalej, 2021*; *Izadi, Akbari & Heydarnoori, 2022*). As mobile devices become more powerful and users demand more visually appealing user interfaces, developers face difficulties during the development of complex visual effects, *i.e.,* animations, media embedding, and lighting. These challenges often lead to display issues, *i.e.,* text overlapping, missing images, and component occlusion, particularly on different types of mobile devices called device fragmentation (*Su et al., 2021*; *Wei, Liu & Cheung, 2016a*). The researchers have identified that most UI issues are caused by various system settings on mobile operating systems, particularly on Android, *e.g.*, different versions of mobile operating systems are running with varying screen resolutions called Android fragmentation (*Wei, Liu & Cheung, 2016a*; *Gomes, da Silva Torres & Côrtes, 2023*). These issues can impact the smooth operation of the application and reduce its accessibility and usability, resulting in a negative user experience and also loss of users as well (*Liu et al., 2020*). The main motivation of this research is to provide automated solutions for such problems. Consequently, the main aim of this research is to detect and fix UI smells to enhance the operation of mobile applications and improve user satisfaction.

All modern applications regardless of desktop or mobile employ a graphic user interface (GUI), also referred as a user interface (UI). This interface provides a visual platform that facilitates communication between software applications and users. When designing a GUI, it is essential to take into account various aspects, such as user interaction, information architecture, and visual design elements (*Miao, 2023*). A well-structured GUI can make an application easy to use, efficient, and reliable, which can have a significant impact on the success of the application (*Coppola, Morisio & Torchiano, 2017*).

One could argue that the principles of design have played a significant role in the development of modern mobile platforms that are highly popular today. For instance, the introduction of the iPhone by Apple in 2007 brought about a revolution in the mobile phone market and had a significant impact on platforms like Android. The iPhone's success was largely attributed to its well-designed user interface, which prioritized natural graphical user interface and multi-touch expressions. Nowadays, the most popular mobile apps on highly competitive app stores like Google Play and App Store focus on simplicity and combine user-friendly designs with engaging user experiences. In fact, given the plethora of apps that offer similar services online today, a platform design and user experience are the distinguishing factors that determine its success or failure (*Yu et al., 2023*; *Abi Kanaan et al., 2023*; *Gomes, da Silva Torres & Côrtes, 2023*).

Smartphone app development usually starts with UI/UX designers creating detailed mock-ups of application displays using various prototyping techniques because an efficient user interface and overall user experience are crucial for the success of the app

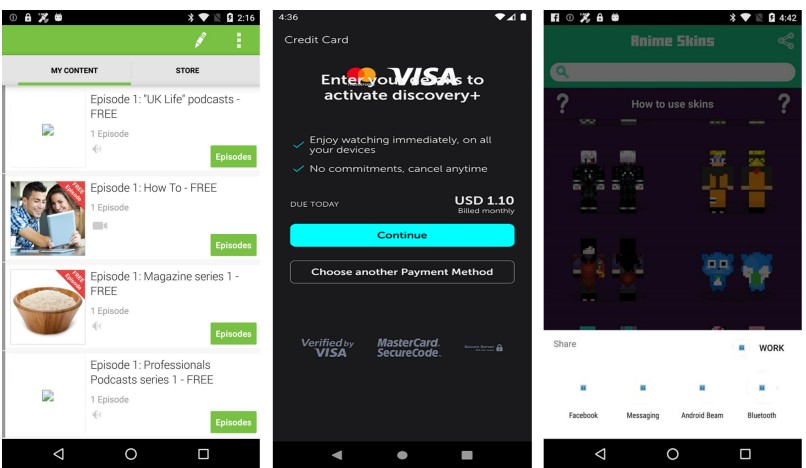

**Figure 1** **Examples of UI display smell.** Image credit: http://interactionmining.org/rico#quickdownloads and Starzplay software house (https://starzplay.com/).

(*Su et al., 2021*; *Su et al., 2017*). Programmers face a significant challenge in implementing more intricate UI and UX effects in mobile GUI design, such as advanced media encoding, animations, lighting, and shadow effects. This often results in various display issues, especially on smartphones, such as text overlapping, missing graphics, and blurry screens, during the UI display process (*Wei, Liu & Cheung, 2016a*) as explained in Fig. 1.

There are over ten different versions of the Android operating system (Android fragmentation) installed on more than 24,000 different mobile devices (device fragmentation) that have distinct screen resolutions. Due to device and Android fragmentation, many display issues arise in the user interface of mobile applications, especially on Android (*Wei, Liu & Cheung, 2016b*; *Ji et al., 2018*). While these graphical user interface (GUI) bugs in mobile apps can not directly cause functionality problems for example app crashing and becoming completely unusable, but impact the app's performance, affect the user experience, and decrease the App users (*White, Fraser & Brown, 2019*; *Degott, Borges Jr & Zeller, 2019*; *Ki et al., 2019*). Therefore, this research aims to pinpoint these UI display problems.

To ensure that the UI of an app is displayed correctly, software companies have two main options: the first one is automated testing and the second one is manual testing. Recently researchers developed many automated testing techniques and tools, all the developed approaches are probability-based, model-based, and learning based (*Li et al., 2019b*; *Pan et al., 2020*) and the second option is to hire many quality assurance engineers (QA) for app testing who can generate test cases that check major functionality of the mobile application. However, it is very difficult to check GUI bugs by automated and manual testing techniques. Human testers can identify only UI display issues that are prominent and can be detected easily (*Yang et al., 2021*; *Chen et al., 2021b*). There are two main drawbacks to these methods. Firstly, it requires a significant amount of manual effort as testers must go through numerous pages in various interactive ways and examine how

the UI appears on distinct operating system versions and different devices with varying screen resolutions and screen sizes. Secondly, it can be challenging to detect minor GUI display issues like text overlap and component occlusion manually. For the resolution of these issues, some software development companies adopt a rapid application development (RAD) approach (*Johan et al., 2023*), which involves creating software and mobile apps very quickly through iterative methods and ongoing customer and user feedback. However, this reactive approach to bug resolution may have already affected users and caused the company to lose market share (*Wetzlmaier & Ramler, 2017*; *Seaman, 1999*; *Jansen, 1998*; *Khan et al., 2019*).

We require a more proactive method to ensure the quality of the UI just before releasing the app. This method should automatically detect potential faults in the GUI and inform developers to address any issues. It would also be beneficial to obtain user feedback for responsive UI assurance. Several studies on automated GUI testing have been conducted (*Ma et al., 2019*; *Wetzlmaier, Ramler & Putschögl, 2016*; *Wetzlmaier & Ramler, 2017*; *Machiry, Tahiliani & Naik, 2013*) by periodically navigating through various sites and performing arbitrary operations (such as clicking, scrolling, and text entry) until crash bugs or explicit exceptions are triggered. *Liu et al. (2020)* presented OwlEye that can detect user interface bugs and localize them. OwlEye adopts a convolutional neural network to detect UI bugs from buggy screenshots of apps. The possibility of extensively deploying the OwlEye technique is severely impacted by two main limitations. The first limitation of OwlEye is that when it is applied for bug localization, the area that is identified by OwlEys is usually too large, making it very difficult to properly guide the developers to fix that UI bug. Secondly, having data collection procedure takes a long time, and it heavily depends on GUI that have display problems. The industry also extensively uses useful automated testing technologies such as Monkey (*Liu et al., 2020*; *Wang et al., 2019*) and Dynodroid (*Wang et al., 2020*). These automated techniques, however, can only identify major crash defects, not UI display problems that escape the system's detection.

To address the limitations of existing studies (*Liu et al., 2022*; *Wang et al., 2022*; *Liu et al., 2020*) this research introduces a novel approach called M-UI-R. This study aims to utilize deep learning models that exploit visual information for the automatic identification and localization of UI display bugs. This approach frames the identification of UI display bugs as an object detection task. To achieve this research objective we utilized the region-based convolutional neural network (Faster-RCNN) model (*Ren et al., 2015*). Faster-RCNN is a deep learning model that is used for object detection which is employed to not just recognize the screenshots with UI display bugs but also precisely locate these visual design bugs within the screenshot. In this way, the proposed research will benefit the developers and testers who can precisely test the graphical user interface of their mobile applications.

The contributions of this study can be highlighted as:

- We present a Mobile-UI-Repair (M-UI-R) an innovative approach utilizing a region-based convolutional neural network (Faster-RCNN) for the detection and localization of mobile UI design smell. As far as our measurements, this is the best approach for smell detection.

- M-UI-R achieves a remarkable average improvement of 26% in precision, 19% in recall, and 21% in f-measure with respect to baseline (*Zhu et al., 2021*). These results indicate that M-UI-R outperforms in the accurate detection and localization of mobile user interface bugs for the maintenance of mobile applications. Additionally, we validate the efficacy of our approach by conducting a survey from eight developers. The feedback from developers indicates that the information acquired through our approach is valuable for enhancing the UI of mobile applications.

The remaining sections of this study are organized as follows: Section 'Literature review' explores related work in the field. Section 'Motivational study' and Section 'UI issues identification and localization' approach provide a detailed explanation of M-UI-R methodology. Section 'Experiment design' and Section 'Results and analysis' describe the experimental design and evaluation process and also present the results. Section 'Discussion' presents a potential discussion of the study. Finally, Section 'Conclusion' concludes the article and outlines potential avenues for future research.

## LITERATURE REVIEW

The mobile GUI serves as a visual connection between users and mobile applications. The quality of the user design interface for mobile apps holds significant importance, especially as technology becomes more pervasive in various aspects of life. Whether in domestic, commercial, or industrial settings, humans rely on numerous software applications to accomplish specific tasks. Therefore, it is crucial for the GUI to be responsive, easily comprehensible, and navigable. Software industries and application developers use many methods and tools to check UI design problems. Therefore many researchers worldwide working to assist designers and mobile application developers on Graphical user interface designing image features and graphical user interface auto code generation by using computer vision based technology (*Zein, Salleh & Grundy, 2016*; *Chen et al., 2019a*; *Zhao et al., 2019*; *Yang et al., 2021*; *Chen et al., 2021a*; *Nguyen & Csallner, 2015*). *Chen et al. (2020a)* propose a novel method utilizing a deep neural network to encode visual and textual data. Their approach aims to generate missing tags for current UI examples, enhancing their discoverability through text-based searches. *Chen et al. (2020c)* investigated the constraints and effective architecture of object detection methods using deep learning for identifying UI components. They introduced a novel top-down approach, progressing from coarse to fine detail, which they integrated with a well-established GUI text deep learning model. *Moran et al. (2018b)* assessed the adherence of implemented GUIs to the original design by employing computer vision techniques to compare image similarities. Subsequently, their follow-up research aimed to identify and summarize GUI alterations in evolving mobile applications (*Moran et al., 2018c*). In contrast to these studies, our focus lies in identifying GUI display anomalies to enhance overall app quality.

The study of *Geel (2019)* is based on the technique developed by the students at the University of Twente enrolled in the Creative Technology (CreaTE) program. Under the curriculum, students of CreaTE need to build a creative and innovative real-life application in their first year. Students can write simple codes but cannot be able to deal with sizeable

programming segments to build interactive GUIs. Processing is a subset language of Java that CreaTE students use to achieve their target. Moreover, many researchers focus on the design smell for processing language code and evaluate its working (*Yu et al., 2023*; *Oliaee et al., 2023*; *Xu, Zhang & Hong, 2022*; *Gomes, Côrtes & Torres, 2022*; *Chen et al., 2021b*; *Li et al., 2019a*; *Mahajan et al., 2018*). A comparison of processing novices and publicly available code was also performed. This comparison leads to the introduction of a constant evaluation tool that automatically detects the design smells from code. It clearly describes how the proposed tool detects the design smells from GUI code and gives suggestions to improve the working of GUI. Recent studies (*Moran et al., 2017*; *Mariani, Pezzè & Zuddas, 2018*; *Denaro et al., 2019*; *Gu et al., 2019*; *Chen et al., 2018*; *Moran et al., 2018a*) noticed the claim of other researchers about the subjectiveness of code smell detection methods that are also very difficult to compile applications code. That studies revolves around three key research questions. The initial inquiry explores the various techniques accessible for detecting code smells. The second question delves into the identification of the most effective strategies from the available resources. Lastly, the third question focuses on visualizing the chosen techniques. The researchers conducted extensive research focusing on code smells detected by different techniques, encompassing search-based, metric-based, symptom-based, visualization-based, probabilistic, cooperative, and manual approaches (*Mirzaei et al., 2016*; *Saga et al., 2022*; *Chen et al., 2019b*; *Moran et al., 2018c*; *Chen et al., 2020b*). The research findings reveal that a substantial proportion of code smell detection algorithms, primarily following search and metrics-based methodologies, lean towards machine learning, as deduced from the comprehensive investigation performed by the authors (*Nayebi, Desharnais & Abran, 2012*; *White, Fraser & Brown, 2019*; *Chen et al., 2019c*; *Feng et al., 2021*). It is shown in the survey results that only 80% of selected 89 techniques can detect code smells (*Ki et al., 2019*; *Di Nucci et al., 2018*; *Pecorelli et al., 2019*). Most of the techniques do not use visualization methods for the detection of UI display issues.

The researchers deny the claim and state that it is not a problematic or trivial task to detect code smells (*Staley, 2015*; *Khan et al., 2023*). However, there are opportunities to improve the detection process and techniques. There should be a way to reduce subjectivity, increase diversity, and create Oracle databases for code smell techniques for data evaluation and visualization. The final line of defense for app quality is manual testing, supplemented by automated GUI testing, especially in identifying usability and accessibility concerns. However, manual testing is labor-intensive, time-consuming, and ineffective due to repetitive tasks and the easy omission of some functionalities. *Liu et al. (2022)* created a tool called NaviDroid to help guide human testers through sequential procedures that are highlighted for more effective and efficient testing. The technique drew inspiration from the video game Candy Crush, where players utilize colorful candies for their moves. For NaviDroid, the two relevant states feature trigger actions situated at the edges of an augmented state transition graph (STG). Leveraging the STG as its foundation, NaviDroid employs dynamic programming to chart the exploration path, enhancing the runtime GUI with visually depicted recommendation movements. This empowers testers to swiftly navigate untested states and reduce redundancy. Automated testing substantiates that

NaviDroid delivers exceptional coverage and efficient route planning but it only cover navigation's bugs of UI. NaviDroid did not include UI aesthetic problems.

Various static bug detection tools serve to ensure the proper functioning of the graphical user interface by identifying code errors, UI bugs, and style inconsistencies (*Zhao et al., 2020*; *Chen et al., 2020b*; *Mahmud, Che & Yang, 2021*). For instance, Android compatibility issue detection is capable of flagging more than 260 types of bugs specific to Android, covering areas such as correctness, performance, security and usability (*Chen et al., 2021b*). Recent studies assist developers in adhering to style conventions and avoiding errors in styling. Multiple surveys (*Zein, Salleh & Grundy, 2016*; *Lämsä, 2017*) have compared various GUI testing tools for Android applications. Some testing approaches concentrate on specific UI concerns such as rendering delays and image loading. They examined potential issues in UI rendering and devised automated methods for their detection. They highlighted challenges in Android app design and implementation arising from the diverse resolutions of mobile devices. Recently, there has been a rise in the use of deep learning-based techniques for automatic GUI testing. Unlike traditional methods that employ dynamic program analysis to explore GUIs, these techniques (*White, Fraser & Brown, 2019*; *Degott, Borges Jr & Zeller, 2019*) utilize computer vision methods to detect GUI components on the screen and determine subsequent actions.

While the GUI testing methods mentioned earlier primarily focus on functional testing, our work centers on non-functional testing specifically, GUI visual bugs that may not lead to app crashes but can significantly impact app usability. The UI display bugs identified by our approach often occur due to compatibility issues (*White, Fraser & Brown, 2019*; *Ki et al., 2019*). The different variations in devices and Android versions have many compatibility issues. Covering all popular contexts manual testing is very costly and also challenging for developers. Furthermore, unlike approaches that rely on static or dynamic code analysis, our method solely necessitates screenshots as input. This characteristic enables our lightweight, computer vision-based approach, which can be applied across various platforms such as Android, iOS, and IoT devices. Web apps are also accessed on devices that have diverse screen widths and present UI bugs. We are also considering these web UI display bugs in our research. In contrast to web apps, Android apps present a wider array of UI display challenges, such as text overlap, missing images, component occlusion and null values often occurring in smaller areas. Consequently, our approach employs computer vision for detecting and localization of user interface display bugs and also test the performance of M-UI-R on different platforms.

## MOTIVATIONAL STUDY

We carried out a thorough study to better understand user-interface display problems or visual smell in real-world situations. In order to improve our method for identifying these flaws in UIs, the study set out to ascertain the prevalence and particular types of UI display difficulties that exist.

## Data collection

We obtained a dataset for our study from three different platforms. The first and main dataset which publicly available for research and development called Rico dataset (http://interactionmining.org/rico#quick-downloads) (*Deka et al., 2017*) which is used for training and testing purposes. The second one is the crowd-sourcing platform Baidu (https://baidu.com). It is also the largest platform where users test various mobile applications. The platform's employees are given jobs to test the applications, and then submit reports outlining their testing and observations made during testing. Between January 2021 and August 2023, we collected 1,443 of these testing tasks for Android mobile applications that were documented in our dataset. The testing assignment required the employees to provide several reports explaining the testing procedure as well as images of the user interface for the program. This dataset was chosen because it contains both screenshots of the user interface and explanations of UI bugs or problems that were discovered during testing. This makes it easier for us to search and examine any issues in the UI having display issues. The third platform is software houses which in minor in count. We consider mobile applications that have more than 5,000 downloads and at least have more than 4.3 star ratings from several areas, including news, entertainment, health, media and finance for example StarzPlay, Perfect Piano, Secure VPN, MediaFire and CEToolbox. We collected 217 UI screenshots from software houses having different UI display issues. In total, we collected 1660 UI screenshots having different UI bugs. These all UI shots were a part of the validation and real-time testing of the M-UI-R approach.

## Categories and visual understandings of UI design smell

There are several categories of UI display issues for example navigation problems, component occlusion, text overlap, missing image, null values, blur screen, background image size that make foreground button text invisible, inappropriate button and icon size. The bugs other than component occlusion, text overlap, null values and missing images can be solved by following the design guidelines of application UI development (*Yang et al., 2021*). However, the targeted UI bugs (component occlusion, missing images, null values and text overlap) can not be identified by following the design rules. The respective UI bugs as explained in Fig. 2 and Table 1. There are multiple reasons for the selection of these bugs for example impact on user experience, frequency of occurrence and also these bugs have negative effects on the usability and functionality of mobile applications. Recent studies (*Wang et al., 2022*; *Ali et al., 2024*) explored UI bugs but they performed analysis on user reviews and feedback available on relevant forums such as Google Play and App Store (https://play.google.com/console/about/reviews/). They used textual features and reported these bugs. These are the factors that motivated us to select these visual design bugs for this research.

The above factors illustrate the severity of GUI bugs and also motivate us to design an automatic approach to successfully identify and locate these UI bugs. A commonly utilized technique for bug detection in mobile applications is to perform program analysis, building of new rules, code rewriting for different platforms such as Android and iOS and customization of code to ensure compatibility on different mobile devices (for example

**Figure 2** **Four different types of UI display smell.** Image credit: http://interactionmining.org/rico# quickdownloads and Starzplay software house (https://starzplay.com/).

**Table 1** **Dataset statistics.** The bold indicates the significance of proposed approach.

| Smell category | Train | Test | Validation |
| --- | --- | --- | --- |
| Text overlap | 6,400 | 800 | 800 |
| Component occlusion | 6,400 | 800 | 800 |
| Missing images | 6,400 | 800 | 800 |
| Null values | 6,400 | 800 | 800 |
| **Total** | **25,600** | **3,200** | **3,200** |

OPPO, Samsung, *etc.*). This process is time-consuming and labor expensive. Especially the task of analyzing all possible display problems and designing their detection rules is not easy.

We examined the screenshots and corresponding JSON files for these four categories of UI bugs. We discovered that text overlap, component occlusion and null value bugs can be identified by analyzing the component information extracted from JSON files (such as component text and coordinates of relevant components). We performed static analysis of JSON file data to identify these three types of bugs using static analysis of XML files. For text overlap analysis we perform analysis of all text views if the coordinates of any two text views overlap then there is a bug same as for component occlusion the coordinates of the component are analyzed if there is an intersection between any two components coordinates then component occlusion bug existed between them. Null value bug identification is very easy, we just examine text of the component if there is "null" text then it is a bug.

Furthermore, the static analysis approaches have many limitations. In certain scenarios, it becomes very challenging to detect these UI bugs just by performing the simple analysis of JSON files. For instance, in the case of component occlusion (as illustrated in Fig. 3), the font size information is not available in the JSON file, making it impossible to detect this bug where the font size is incompletely displayed in an EditText. Additionally, as depicted in Fig. 3 again, elements such as dialog box, toolbar and spinner may overlay the component, creating interference in the detection process. Similarly, for text overlap issues, there is potential interference from component occlusion. Regarding null value problems, the text extraction process may yield numerous null values, many of which stem from

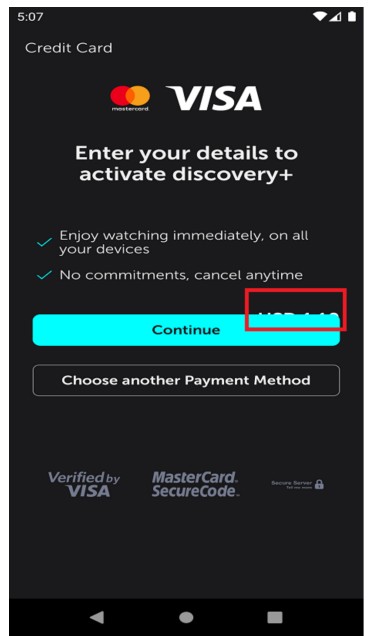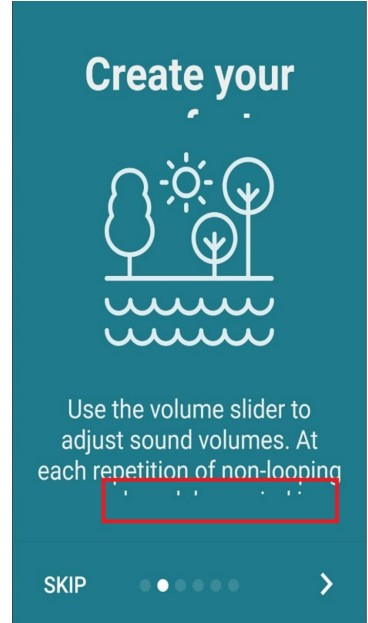

**Figure 3** **Examples of undetectable UI display smell.** Image credit: http://interactionmining.org/rico#quickdownloads and Starzplay software house (https://starzplay.com/).

issues during JSON file retrieval rather than actual UI display problems. This introduces considerable noise into the bug detection process.

Considering this perspective, it is valuable to implement efficient, fast and versatile method for detecting UI display bugs. Drawing inspiration from the fact that humans can recognize these display issues visually, we suggest identifying these problematic screenshots using a visual understanding technique that mimics the human visual system. Since UI screenshots can be obtained easily (either manually or automatically) and generally do not exhibit significant variations across apps from various platforms or devices, our computer vision-based approach offers greater flexibility and ease of implementation.

## UI ISSUES IDENTIFICATION AND LOCALIZATION APPROACH

This article introduces the concept of Mobile-UI-Repair (M-UI-R) which is designed to identify and pinpoint UI display problems by using screenshots of the tested mobile applications. M-UI-R's function is to automatically detect and identify these issues and determine their location on the user interface of the application. When the user presents a UI screenshot to M-UI-R seamlessly combines detection and localization capabilities. By employing visual comprehension, it identifies screenshots linked to UI display problems and marks the specific problem areas using bounding boxes. This aids developers in efficiently addressing and resolving these bugs.

Since visual cues often reveal UI display problems, we have chosen to utilize the region-based convolutional neural network (Faster-RCNN) framework introduced in 2015 (*Ren et al., 2015*). This is a deep learning-based object detection framework. That framework

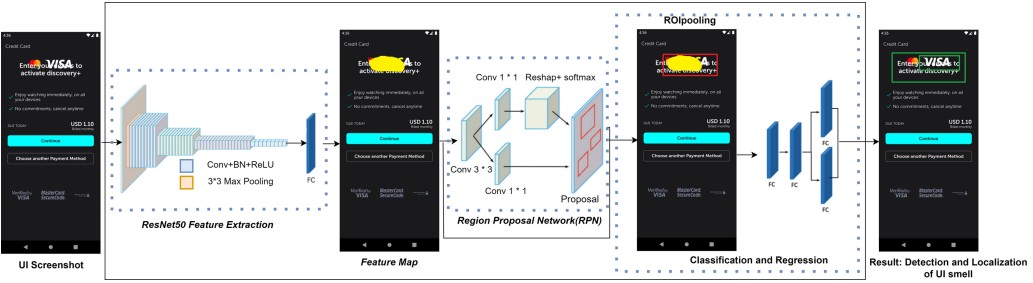

**Figure 4** UI smell detection and localization approach.

particularly demonstrated its efficiency in detecting objects within the realm of computer vision. The architecture of our proposed M-UI-R object detection model is presented in Fig. 4. It has three key components: the first one is feature extraction by using a network (ResNet50) which is a deep learning model used for image classification (*Koonce, 2021*), the second component is regional proposal network (RPN) module that is used to generate region proposals or candidate bounding boxes for objects in an image, and the third and final module is Region of Interest (ROI) pooling module. Faster-RCNN utilizes this ROI pooling module to extract fixed-size features from feature maps that are already extracted by convolutional layers. These components resize the given screenshot into a specific size and dimension represented as "width(w) x height(h)" and then standardize the image. After that, these screenshots are input into the convolutional neural network ResNet50 (*Jian et al., 2016*). The convolutional layer comprises learnable filters that function as parameters. The major objective of these convolutional operations is to extract the distinct features from the input (a process known as feature extraction). Following the convolutional layer, the UI screenshots are transformed into a feature set called a feature graph. As the network's depth increases, the accuracy increases in the beginning and then rapidly declines. This phenomenon often leads to issues of exploding gradients and degradation.

This issue arises due to the gradients moving backward through the layers, and through repeated multiplicative operations, the gradient can become extremely small. Consequently, as the network's depth increases, its performance tends to reach a saturation point or even decline precipitously. To address this challenge, ResNet50 introduces the notion of residual errors. It uses a deep residual learning approach to overcome the degradation issue. ResNet50 explicitly aligns these layers using a residual mapping technique rather than each layer to directly match a desired mapping. We use ResNet50 to derive the feature map, which is then input into the Region Proposal Network (RPN) module. The RPN receives a feature map as input, which may be of any size, and outputs three rectangular object recommendations, each of which is accompanied by a score indicating how likely it is to contain an object. A sliding 3x3 size window is used to navigate the feature map in order to do this.

A series of nine anchors are created for the center of each window during the traversal process. This takes into consideration various ratios and scales, specifically 1:2, 1:1, and 2:1. Subsequently, each anchor undergoes classification as foreground or background through

a fully connected technique, and initial estimates for bounding boxes are also generated. These anchors are then refined using these bounding box estimations, resulting in more precise and accurate suggestions.

Next, we direct these refined anchors to the ROI pooling layer for the computation of proposal feature maps. This is achieved by utilizing the feature map obtained during the feature extraction phase along with the proposals generated by the RPN module. Subsequently, these proposal feature maps are directed to the categorization module. Employing fully connected neural networks (FC) in conjunction with a softmax layer, this module allocates each recommendation to a specific category. These categories encompass component occlusion, missing images, text overlap, and missing values. The outcome is a probability vector that illustrates the probability of each category occurrence.

Simultaneously, employing the bounding box expression once more, we determine the positional offset of each proposal. This offset information proves valuable in refining the accuracy of target detection frames, allowing for more precise localization of the identified issues.

## EXPERIMENT DESIGN

### Research questions
- Research question 1: How effective is our M-UI-R approach in the detection of UI display bugs?
- Research question 2: How effective is our M-UI-R approach in the localization of UI display bugs?
- Research question 3: Is the information extracted by using our M-UI-R approach useful for improving mobile applications?

### Experimental setup
Since our Proposed approach "M-UI-R" is fully automated and we used an already published dataset called Rico dataset (*Deka et al., 2017*). We also collected data from crowd-sourcing platforms and software houses for real-time analysis of the proposed approach. To create a balanced dataset we used an equal number of images (screenshots) from each category. To make our approach effective we chose 8,000 screens from every category and in total we used 32,000 screens for analysis, as shown in Table 1. Among these 32,000 half are positive and half are negative, positive means without bugs and negative means with bugs. we used 80% data as a training dataset and 20% for testing and validation.

In addition to checking the real-time performance of our approach, we employed on real-time dataset obtained from crowd-testing apps. This dataset includes 800 screenshots including 400 buggy and 400 without bugs for testing. We exploited 5-fold cross-validation.

### Baselines
We conducted a comparison between our M-UI-R approach and five baseline methods that combine machine learning and deep learning techniques. This was done to emphasize the advantages of our proposed approach. Among the baselines, three employ machine learning techniques to extract visual effects from the screenshots and subsequently apply

learning methods for categorization. The remaining two baseline approaches directly employ deep learning concepts, using artificial neural networks for classification tasks. Our primary focus is to elucidate the feature extraction procedure utilized in machine learning-based techniques.

The scale invariant feature transform (SIFT) (*Lowe, 2004*): represents a widely used method for extracting features. It identifies and characterizes local features within an image. This technique isolates intriguing points on an object to generate a feature description that remains unaffected by uniform scaling, orientation variations, and changes in illumination.

Speeded-up robust features (SURF) (*Bay, Tuytelaars & Van Gool, 2006*): represents an enhancement of the SIFT technique. Within SURF, the Hessian blob detector is approximated using integer values. This approximation can be achieved with the assistance of a precomputed integral image, requiring only three integer operations for its computation.

The oriented fast and rotated brief (ORB) (*Rublee et al., 2011*): this technique characterized by its speed in both feature point extraction and description, is detailed in the ORB article. ORB builds upon the foundation of BRIEF, a swift binary descriptor. Notably, ORB incorporates rotation invariance and robustness against noise interference. Following this, we apply four widely used machine learning methods utilizing these extracted features: support vector machine (SVM) (*Kotsiantis, Zaharakis & Pintelas, 2007*), k-nearest neighbor (KNN) (*Sumiran, 2018*), naive Bayes (NB) (*Kotsiantis, Zaharakis & Pintelas, 2007*), and random forests (RF) (*Breiman, 2001*). These methods are employed to classify screenshots depicting UI display issues.

MLP (*Zhu et al., 2021*; *Driss et al., 2017*): the multilayer perceptron (MLP) forms a feedforward artificial neural network. This network architecture comprises an input layer, hidden layers, and an output layer. Neurons within this structure operate with a non-linear activation function similar to the rectified linear unit (ReLU). During the training process, the connection weights are adjusted while considering the output's deviation from the ground truth. In our approach, we employ an eight-layer neural network with the following neuron counts in each layer: 190, 190, 128, 128, 64, 64, 32, and 2.

OwlEye (*Liu et al., 2020*): OwlEye utilizes a convolutional neural network (CNN) as its foundation to recognize the screenshots containing UI display bugs. Additionally, it employs Gradient weighted Class Activation Mapping function (Grad-CAM) to pinpoint the areas where UI design issues occur.

## Evaluation metrics

We employ three commonly used evaluation metrics: precision, recall, and F1-score, to assess the effectiveness of our proposed research in addressing the challenges outlined in RQ1. These metrics find widespread application in the domains of pattern recognition and image classification (*Zein, Salleh & Grundy, 2016*; *Chen et al., 2019a*). A higher value always indicates good performance.

Precision and recall are typically calculated by aggregating the counts of true positives (TP), true negatives (TN), false positives (FP), and false negatives (FN). Within the context of issue identification, TP corresponds to correctly labeled as buggy during prediction, FN

to inadvertently mislabeled non-bug images as buggy, TN to correctly identified normal screenshots, and FP to incorrectly labeled as non-buggy screenshots.

Precision signifies the ratio of correctly predicted UI display issue-containing screenshots to all the screenshots predicted as buggy:

$$Precision = \frac{TP}{TP + FP}. \tag{1}$$

Recall denotes the ratio of accurately predicted buggy screenshots from all the screenshots that genuinely exhibit UI display issues.

$$Recall = \frac{TP}{TP + FN}. \tag{2}$$

The F1-score (also known as F-measure or F1) represents the harmonic average of precision and recall, effectively combining both of the aforementioned metrics.

$$F - measure = \frac{2 \times Precision \times Reccall}{Precision + Recall}. \tag{3}$$

For issue localization, we utilize two frequently employed evaluation metrics such as average precision (AP) and average recall (AR) which are commonly used in the context of object detection (*Ren et al., 2015*). These metrics are applied to assess the efficacy of our proposed approach in addressing the localization challenges described in RQ2. AP and AR offer a more precise and comprehensive evaluation of M-UI-R localization capabilities. Larger values consistently denote improved performance. While precision and recall are analogous in image categorization, AP and AR serve distinct evaluation purposes.

We start by selecting prediction boxes with confidence values exceeding 0.55 (*Ren et al., 2015*). Subsequently, we assess the overlap between the predicted UI visual smelly region and the actual smelly region by using the intersection over union (IoU) ratio. This ratio is calculated using the following formula: IoU equals the intersection of the predicted UI smelly region and the actual UI smelly region divided by the union of these two. IoU effectively addresses coverage issues. The TP refer to the count of identification boxes with an IoU of 0.55 or higher. The same false positives encompass detection boxes, including redundant ones identified within the same ground truth box, that have an IoU less than 0.55. The term FN represents the number of ground truth boxes that are not detected.

## RESULTS AND ANALYSIS

### UI smell detection performance (research question 1)

Table 2 presents the initial assessment of our proposed Mobile-UI-Repair (M-UI-R) approach, both in terms of fault detection and its performance across four distinct types of display issues of mobile UI within the training dataset and the real-world dataset (collected from crowd-sourcing and software houses). Figure 5 represents the precision and recall performance of UI bug detection on different size of training data. M-UI-R attains an average precision (AP) of 0.877 in the training dataset, denoting that 87.7% (702 out of 800) of the predicted problematic screenshots genuinely exhibit UI display faults. Furthermore, the average recall (AR) stands at 0.865, signifying that M-UI-R adeptly

**Table 2  UI smell detection performance (In %).** The bold indicates the significance of proposed approach.

| Smell category | Precision | Recall | F-measure |
| --- | --- | --- | --- |
| Text overlap | 91.9 | 83.7 | 87.6 |
| Component occlusion | 82.5 | 77.8 | 80.1 |
| Missing images | 89.2 | 90.7 | 89.9 |
| Null values | 87.3 | 93.8 | 90.4 |
| **Average** | **87.7** | **86.5** | **87.1** |

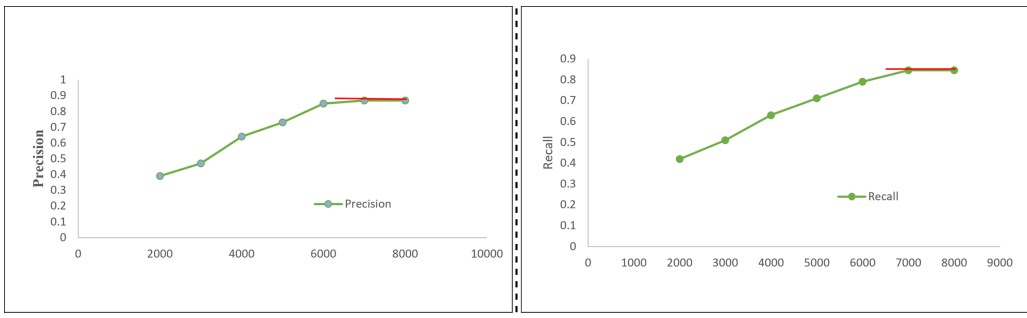

**Figure 5  Precision and recall result on different training dataset.**

identifies 86.5% (692 out of 800) defective screenshots. In the real-world dataset, M-UI-R yields an average precision of 0.856. This signifies that, on average, 85.6% percent (186 out of 217) of the identified screenshots with UI display faults do indeed exhibit such issues. The average recall for the real-world dataset stands at 0.843, highlighting that M-UI-R effectively detects 84.3% (183 out of 217) of the problematic screenshots.

Even though M-UI-R underwent training due to a very small dataset that's why its average precision and recall on the real dataset were slightly lower, measured at 0.021 and 0.022, respectively, compared to those in the training dataset. This observation serves to validate the effectiveness of our M-UI-R system.

We shift our attention to the upper section of Table 2, where we delve into the performance metrics for each distinct category of UI display difficulties. It is worth highlighting that the precision and recall measurements for all four categories are quite commendable. While the lowest recorded precision is 82.5% and the recall value is 77.8%, respectively, the highest precision value is 91.9% and the recall value is 93.8%. Same like the "null value" category of UI smell stands out with the highest F-measure, showcasing a remarkable balance between precision (0.87) and recall (0.93). This observation could be attributed to the recurring nature of null value issues in screenshots, which often makes identifying the problematic areas relatively straightforward. In contrast, the "component occlusion" category of UI smell attained the lowest value of precision and F-measure respectively. This distinction is due to the diversified pattern of each category of bug, as well as the relatively smaller size of the buggy area for example in the case of component occlusion the buggy area is only 8.5%. While in case of missing images, it is visible and can

be identified easily. In future work, we will try to improve the identification and localization performance with image magnification.

Furthermore, a detailed analysis of screenshots that were erroneously labeled as bug-free has been carried out, and illustrative instances are explained in motivational study section. These images often consist of very few problematic regions that are challenging to discern, even for human observers. Our upcoming efforts will concentrate on incorporating attention mechanisms and image magnification techniques to enhance the detection performance for such types of screenshots.

## Issue localization performance (research question 2)

Illustrations showcasing our problem localization are displayed in Fig. 6, concentrating on the precise positions of the identified bugs. OwlEye (Liu et al., 2020) localization outcomes are visualized as heat maps, prompting us to employ image binarization for delineating the bounding box encompassing the highlighted region within the heat map. This facilitates a direct comparison of its efficacy with the novel approach proposed by M-UI-R. The efficiency of M-UI-R problem localization is outlined in Table 3. The average values for issues localization for real and testing datasets are AP (average precision) and AR (average recall) achieved by M-UI-R are 71.5% and 69.7%, respectively.

Let us now thoroughly examine the performance of problem localization for each grouping of UI display problems. Across all four types, both AP and AR demonstrate notably high values. Within the real dataset, the lowest recorded average precision and average recall stand at 67.3% and 65.5%, respectively. Notably, the "missing image" category showcases the most outstanding performance, displaying statistically significant AP (75.7%) and AR (74.5%) values. This could be attributed to the substantial and conspicuous buggy regions present in screenshots with missing image problems. Conversely, the "component occlusion" category displays comparatively lower performance, with AP and AR values of 71.5% and 69.5%, respectively. This discrepancy could be attributed to the significantly smaller buggy regions found within this category.

As depicted in Fig. 6, the red bounding box represents the ground truth data, whereas the same bounding box represents the prediction. While the projected bounding box size might be slightly larger than the actual one, leading to an IoU value below 0.55, the indicated buggy area is mostly accurate. This offers valuable insights to developers.

It is important to note that the differing assessment criteria significantly contribute to baseline considerably poorer AP and AR outcomes. In prior studies, participants were tasked with evaluating whether the localized region by OwlEye (Liu et al., 2020) coincided with the actual issue area, serving as a measure of UI display issue localization. The highlighted problematic area was required to share at least 45% of its similarities with the real issue region, whereas AP and AR metrics necessitate an IoU area exceeding 0.55. Notably, we observe three instances where AP/AR metrics suggest lower localization precision, yet human evaluation indicates strong performance. In the initial case, OwlEye highlights only one out of three potential issue sites, while M-UI-R accurately identifies all three, yielding superior AP/AR metrics compared to OwlEye. In the second scenario, there is localization noise, and OwlEye incorrectly designates a different location, leading to decreased AP/AR

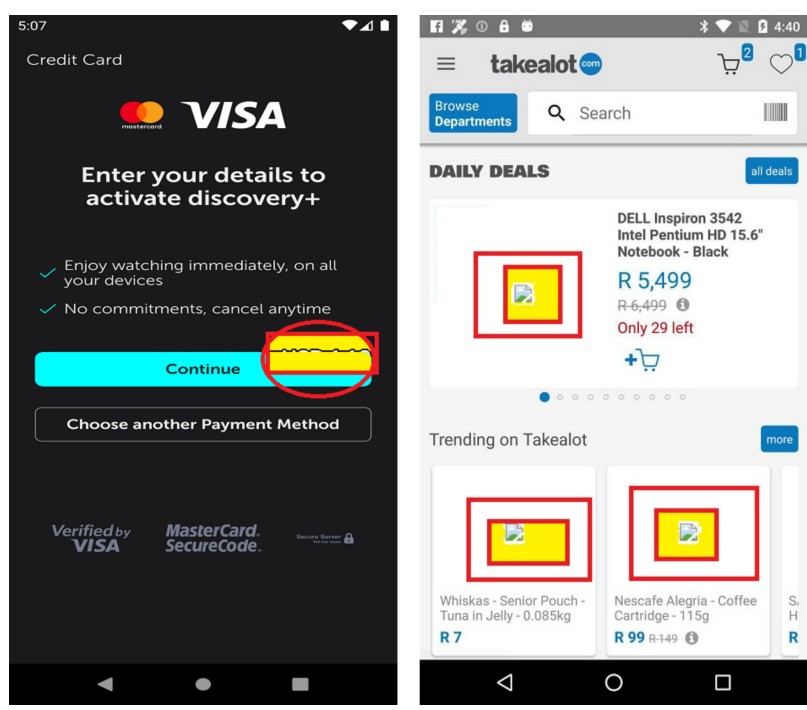

**Figure 6** Localization of UI display smell. Image credit: http://interactionmining.org/rico#quickdownloads and Starzplay software house (https://starzplay.com/).

**Table 3** Performance of issues localization (In %). The bold indicates the significance of proposed approach.

| Smell category | OwlEye | | M-UI-R | |
|---|---|---|---|---|
| | Avg. Precision | Avg. Recall | Avg. Precision | Avg. Recall |
| Text overlap | 41.5 | 49.5 | 72.5 | 67.6 |
| Component occlusion | 32.5 | 37.3 | 67.3 | 65.5 |
| Missing images | 41.7 | 47.7 | 75.7 | 74.5 |
| Null values | 39.3 | 45.5 | 70.5 | 67.2 |
| **Average** | **38.7** | **45.0** | **71.5** | **69.7** |

values compared to M-UI-R. The third situation involves localization drift, where OwlEye highlighted area is not perfectly aligned with the target, while M-UI-R adeptly identifies the entire problem area.

## Performance comparison with baselines

The comparison of performance of our proposed approach with baselines is presented in Table 4. The initial focus is on opposing M-UI-R against the five baselines explained in the experimental setup section. The results indicate the superiority of our proposed approach M-UI-R over the five baselines. Specifically, M-UI-R boasts a precision 28.8% higher and a recall 21.7% higher than the baseline article multilayer perceptron (*Zhu et al., 2021*) and also outperform 10.8% in precision and 11.7% in recall then OwlEye (*Liu et al., 2020*).

**Table 4   UI smell identification performance comparison with baseline (In %).** The bold indicates the significance of proposed approach.

| Method | Training data | | | Real data | | |
|---|---|---|---|---|---|---|
| | Precision | Recall | F-measure | Precision | Recall | F-measure |
| SIFT-SVM | 53 | 51 | 52 | 50 | 49 | 50 |
| SIFT-NB | 56 | 57 | 56 | 52 | 51 | 51 |
| SIFT-RF | 56 | 57 | 56 | 51 | 52 | 51 |
| SURF-KNN | 56 | 56 | 57 | 53 | 52 | 53 |
| SURF-NB | 59 | 59 | 58 | 54 | 53 | 54 |
| SURF-RF | 58 | 59 | 58 | 53 | 54 | 53 |
| ORB-SVM | 56 | 54 | 55 | 51 | 52 | 52 |
| ORB-KNN | 57 | 56 | 56 | 52 | 53 | 53 |
| ORB-NB | 59 | 58 | 58 | 55 | 54 | 54 |
| ORB-RF | 58 | 59 | 58 | 53 | 54 | 54 |
| MLP | 61 | 67 | 64 | 55 | 54 | 54 |
| OwlEye | 79 | 77 | 78 | 76 | 74 | 75 |
| **M-UI-R** | **89.8** | **88.7** | **89.2** | **85.6** | **84.3** | **85.4** |

The other three machine learning baselines demonstrate their performance in terms of precision and recall and f-measure as shown in Table 4. M-UI-R stand superior at least 30% in precision and more than 30% in recall among all three machine learning based baselines (SIFT, SURF and ORF). We reaffirm the effectiveness of M-UI-R, underscoring its adeptness in identifying problematic screenshots from a pool of candidates. Notably, multilayer perceptron (MLP) achieves the highest precision, F-measures, and recall from all the baselines, suggesting the superior performance of this deep learning approach in recognizing faulty screenshots.

Furthermore, a comparison between the performance of the freshly designed model within M-UI-R and the prior model used in OwlEye, utilizing the same data for training and testing, is conducted. The findings demonstrate that the M-UI-R model attains enhanced performance, achieving a precision of 87.7% *versus* 79.0% and a recall of 86.5% *versus* 77.0%. The evident improvement in precision and recall within M-UI-R can potentially be attributed to the detection of object assignment, which offers more precise bounding boxes for UI display problems during model training. This refined data provision facilitates the model in learning relevant features more effectively. In contrast, other baseline methods rely solely on category labels for their training data. M-UI-R robust performance in issue detection, coupled with its capacity to adapt without requiring manual data labeling, makes it particularly adept at accommodating the varied styles of Android UI.

## Usefulness evaluation (research question 3)

To comprehensively assess the effectiveness of our M-UI-R solution, we carried out a random sampling of 3,200 Android applications, 1,600 from FDroid6 and 1,600 from the Google Play store. These applications included newer releases of mobile applications for 2019-2023. Notably, none of these apps were included in our training dataset.

We examined these mobile applications and captured screenshots of their UI pages using DroidBot, a well-known and lightweight Android test input generator (*Li et al., 2017*). Out of the 3,200 applications, only 40% (1,280 out of 3,200) generated multiple screenshots, while 60% (1,920 out of 3,200) were successfully evaluated using DroidBot. The remaining portion of applications, for which DroidBot could not proceed further, typically required user registration or authentication steps. On average, eighteen screenshots were gathered for each of the 1,280 applications. Subsequently, we fed these screenshots into M-UI-R to identify potential UI display issues. Upon detecting an issue, we created a comprehensive bug report outlining the problem and included the relevant problematic UI screenshot. The development teams of these applications were then notified of these bug reports *via* email or issue tracking systems. M-UI-R identifies and provides a comprehensive listing of all the detected issues, and this approach offers supplementary detailed information regarding the identified defects. A total of 97 UI display bugs were detected in F-Droid applications, with 27 of them already fixed and 18 having received acknowledgments from developers. Similarly, in the case of the Google Play store, 76 UI display issues were uncovered, resulting in 37 fixes and nine developer acknowledgments. The rectification of these reported imperfections stands as tangible evidence of the practicality and effectiveness of our proposed approach for detecting UI display problems.

The findings reveal that M-UI-RT not only identifies 79 additional issues that OwlEye misses but also encompasses all the issues detected by OwlEye. The precision of M-UI-R stands at 87% (171 out of 197), marking a 16% improvement over OwlEye precision (87% compared to 71%). M-UI-R demonstrates superior performance in detecting problems with relatively smaller problematic regions. The significance of UI display issues on user experience is further highlighted by developers validation of the issue reports provided by M-UI-R, underscoring the crucial role of M-UI-R in identifying issues.

We also conducted the survey from eight app developers to analyze whether the information extracted by our method can provide help for improving the UI of apps. Initially, we selected 10 buggy screenshots from each category of bugs randomly and applied M-UI-R to identify and localize UI display issues. Subsequently, we presented the buggy screenshots along with the acquired information to developers and pose them questions outlined in Table 5. Specifically, the research question (RQ1) assesses the identification of UI display issues, while the research question (RQ2) evaluates the localization of UI display issues. Each question was accompanied by five options (1 strongly disagree, 2 disagree, 3 neither, 4 agree, and 5 strongly agree).

The developer's feedback is also explained in Table 5. Examining the response to RQ1, it is evident that seven out of eight developers agreed that M-UI-R is beneficial for the identification of UI display issues, with six of them expressing strong agreement regarding the usefulness of the information. Only one developer expressed a conservative response. Similarly, for RQ2, positive feedback was received from six developers out of eight who agreed with the usability of M-UI-R for issue localization. When developers watch the bugs that are detected by M-UI-R they said that these UI bugs have a lot of impact on usability of mobile apps and need to be fixed immediately. These suggestions from developers also validate the usability of M-UI-R.

**Table 5  Survey questions and results.**

| Questions | Strongly disagree | Disagree | Neither | Agree | Strongly agree | Total |
|---|---|---|---|---|---|---|
| **RQ1**: Do you think that M-UI-R is more effective in the detection of UI display issues? | 0 | 0 | 1 | 1 | 6 | 8 |
| **RQ2**: Do you think that M-UI-R is more effective in the localization of UI display issues? | 0 | 0 | 2 | 1 | 5 | 8 |

## DISCUSSION

The generality of M-UI-R approach: The predominant portion of existing research on GUI bug detection (*Li et al., 2017*; *Zein, Salleh & Grundy, 2016*; *Zhao et al., 2019*) tends to be confined to a singular platform, such as Android, thereby constraining its real-world applicability. In contrast, our M-UI-R approach primarily focuses on pinpointing UI display faults. Considering the relatively minimal divergence between screenshots obtained from various systems (such as Android, iOS, Mac OS, and Windows), our methodology can be extended to encompass the identification of UI display issues across diverse platforms.

A limited-scale experiment was conducted on three other prominent platforms such as Windows, Mac operating system and on iOS as shown in Fig. 7. In this experiment, 81 screenshots depicting UI display issues were captured from widely utilized applications and subjected to analysis. The outcomes reveal that our M-UI-R technique accurately identifies 71 out of 81 problematic screenshots, accounting for 87.5% of them. This outcome demonstrates the versatility of the M-UI-R approach and we have a plan for more comprehensive future experiments. Another benefit of M-UI-R is its capability to detect UI display issues across various display languages used in applications. The testing data for the experiment involves screenshots in English and Chinese. The proposed approach successfully identifies UI bugs in different languages as shown in Fig. 7. This illustrates the generalizability and applicability of our approach across different languages.

Enhancements in automated testing tools hold promise for improvement: Findings from RQ3 highlighted the effectiveness of M-UI-R in combination with automated testing utilities such as DroidBot (*Li et al., 2017*). However, multiple Android applications did not permit screenshots due to their security policy. These applications can not be executed with the help of DroidBot and some of them can be accessed with a single screenshot due to limitations. These are the factors that restrict the comprehensive exploration of UI screenshots. The potential of M-UI-R to detect UI display issues in real-world scenarios could undoubtedly be amplified through its integration with a more proficient automated testing tool.

### Threats to validity

We acknowledge potential threats to the construct validity of our proposed approach, particularly concerning the choice of evaluation metrics. The metrics we employ are precision, recall, and F-measure which are widely used in the research community. However, it is important to recognize that the extensive reliance on these metrics may introduce limitations in terms of construct validity.

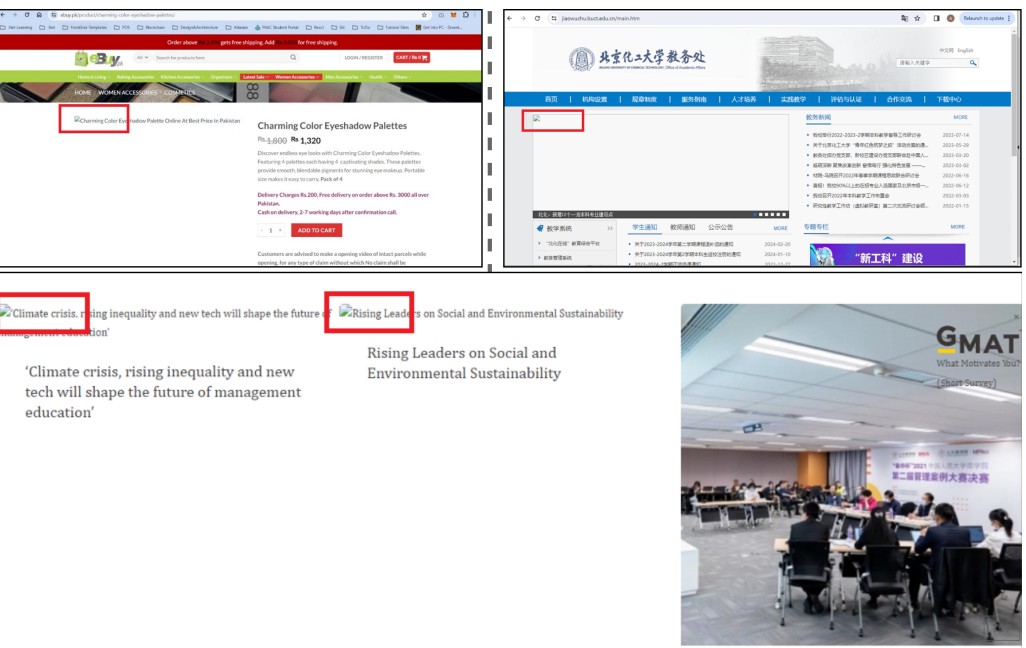

**Figure 7** **Examples of UI display bugs on different platforms.** Image credit: Baidu (https://www.baidu.com).

Furthermore, we are aware of the susceptibility of classification algorithms to validity threats stemming from parameter values. To address this concern, we conducted experiments to determine variations in hyperparameter settings such as batch size or learning rate rather than relying on default values. Nevertheless, it should be noted that adjustments to these parameters could potentially impact the results.

The use of the Faster-RCNN model for the identification and localization of UI bugs introduces a potential threat to construct validity. While there are alternative tools available, we opted for Faster-RCNN due to its superior performance compared to other models available at the time. Nonetheless, it is crucial to acknowledge that the absence of comprehensive object detection tools for UI bug detection could affect the overall performance of our proposed approach.

Internal validity concerns arise from the implementation of our approach. To mitigate this, we perform cross-checks to ensure the accuracy of our methodology. However, there remains a possibility that some errors may have been overlooked.

In terms of external validity, there are considerations regarding the generalization of our proposed approach. Although our analysis is confined to mobile application UI bugs, our approach could be applicable to all platforms for the detection of UI bugs.

Lastly, the limited number of projects in our study poses an external validity threat. Deep learning algorithms often require fine-tuning parameters and substantial training data to achieve optimal performance. The restricted number of projects may therefore limit the generalizability of our results and our ability to fully explore. We would also discuss the

different categories of applications, like testing a streaming application might be different with respect to an accountability, utility, social or chat application.

A limitation of our proposed study is that it can not identify the design smell related to the style of UI, layout bugs of UI, button size, typography, iconography and navigation design flaws. These style bugs also have a great impact on the user experience of mobile applications. We will consider these bugs in our future research.

## CONCLUSION

Enhancing the quality of mobile applications, particularly proactively, holds significant value. The core aim of this study is to automatically detect UI display issues in screenshots generated during automated testing. Through the successful identification and validation of 79 previously unnoticed UI display bugs in renowned Android apps, the newly introduced M-UI-R technique has demonstrated its effectiveness in real-world scenarios. Furthermore, M-UI-R outperforms the leading deep learning-based baseline (MLP), showcasing a remarkable enhancement of more than 19% in recall and over 26% in precision. Our pioneering contribution encompasses an exhaustive examination of UI display challenges in genuine mobile applications, along with the assembly of a substantial assortment of app UIs featuring display issues, intended to facilitate further research.

Moving forward, our efforts will continue to focus on enhancing our model to achieve improved classification performance. In addition to detecting display issues, our upcoming work will delve into pinpointing the underlying causes of these problems. Subsequently, we intend to develop a suite of tools designed to recommend patches to developers, aiding in the resolution of display-related bugs.

## ACKNOWLEDGEMENTS

We are thankful to the anonymous reviewers for their constructive feedback and valuable suggestions, which greatly enhanced the quality of this article.

### Funding
This research was supported by the Princess Nourah bint Abdulrahman University Researchers, Project number (PNURSP2024R409), Princess Nourah bint Abdulrahman University, Riyadh, Saudi Arabia. The funders had no role in study design, data collection and analysis, decision to publish, or preparation of the manuscript.

### Grant Disclosures
The following grant information was disclosed by the authors:
The Princess Nourah bint Abdulrahman University Researchers, Princess Nourah bint Abdulrahman University, Riyadh, Saudi Arabia: PNURSP2024R409.

### Competing Interests
The authors declare there are no competing interests.

## Author Contributions

- Asif Ali conceived and designed the experiments, performed the experiments, analyzed the data, performed the computation work, prepared figures and/or tables, authored or reviewed drafts of the article, and approved the final draft.
- Yuanqing Xia conceived and designed the experiments, performed the experiments, analyzed the data, performed the computation work, prepared figures and/or tables, authored or reviewed drafts of the article, and approved the final draft.
- Qamar Navid performed the experiments, authored or reviewed drafts of the article, and approved the final draft.
- Zohaib Ahmad Khan conceived and designed the experiments, analyzed the data, authored or reviewed drafts of the article, and approved the final draft.
- Javed Ali Khan analyzed the data, performed the computation work, prepared figures and/or tables, authored or reviewed drafts of the article, and approved the final draft.
- Eman Abdullah Aldakheel performed the experiments, analyzed the data, performed the computation work, prepared figures and/or tables, and approved the final draft.
- Doaa Khafaga conceived and designed the experiments, performed the computation work, prepared figures and/or tables, and approved the final draft.

## Data Availability

The code and data are available in the Supplemental File. The Rico dataset is available at http://www.interactionmining.org/rico.html.

## Supplemental Information

Supplemental information for this article can be found online at http://dx.doi.org/10.7717/peerj-cs.2028#supplemental-information.

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
