# Peer review of "Mobile-UI-Repair: a deep learning based UI smell detection technique for mobile user interface"

_PeerJ Computer Science, doi:10.7717/peerj-cs.2028_

## Round 0.1 · original submission · Major Revisions

Both reviewers suggest the manuscript could benefit significantly from thorough proofreading to address language errors, odd hyphenation, and clarity issues. They recommend a more precise structure separating results and discussion sections, an expanded discussion on the paper's contributions compared to existing literature, and a section addressing potential threats to validity. This would not only improve the readability and coherence of the paper but also strengthen the argument for the proposed technique's significance in the field of GUI design assessment.

**Language Note:** The review process has identified that the English language must be improved. PeerJ can provide language editing services - please contact us at [email protected] for pricing (be sure to provide your manuscript number and title). Alternatively, you should make your own arrangements to improve the language quality and provide details in your response letter. – PeerJ Staff

Reviewer 1 ·

Basic reporting

The paper proposes a UI bug detection and identification technique called Mobile-UI-Repair(M-UI-R) which is capable of recognizing graphical user interfaces (GUIs) display issues and identifying the specific location of the bug within the GUI. The paper is well-written and nicely presented.

Experimental design

Comprehensive experiments are conducted to support the results.

Validity of the findings

Results capable of recognizing graphical user interfaces (GUIs) display issues and identifying the specific location of the bug within the GUI.

Additional comments

Improvement areas:

1. In the abstract “Although several approaches have been proposed, they require significant
performance improvement”. Need to describe those requirements in the introduction section.
2. In Section VI, the authors need to cite the paper for owlEye wherever they are comparing their method's performance with owlEye.
3. There should be a clearer distinction between Section 6 and 7. The results should be presented in Section 6 whereas the results should be discussed in Section 7. Section 7 should discuss the results separately for each research question.
4. The paper mentioned conducting a survey by 8 developers in the abstract which has not been further discussed or presented in the later part of the paper.
5. There are many language-related errors. For example: "The authors of the study [43]–[46]". Therefore, the manuscript needs to be proofread.
6. The overall presentation of the paper is good. Few issues can be considered for improvement.
• In multiple places hyphenation seems odd
• In the whole manuscript, there are lots of convincing issues, which makes it
difficult to read please describe them Properly.
• Consider to refine the content.
7. Discussion should be extended. Comparisons of results against previous papers could highlight the significance of this paper.

Reviewer 2 ·

Basic reporting

The paper describes an interesting approach to assess GUI from the design point of view. The suggested technique can find flaws and display errors inside a mobile user interface. Authors compared their approach to others existing in the literature, pointing out that their work performs better than the other approaches in the selected dataset. Overall the paper is well-written, although some aspects might be improved in order to achieve higher clarity and inform the reader on the experimental design choice.

I do not totally agree with the claim "While these problems do not hinder the software’s functionality, but impact the app’s performance, affect the user experience, and decrease the App users" I'm not sure that those kinds of errors result in a decrease of app users. Please, provide a reference to support that.
Also, I don't agree that companies have two options (line 81). Maybe those are the most popular, but those are not the only options they have, please discuss better or provide a reference to support that claim.

I found the introduction quite redundant in some point, insisting in really basic concepts repeating in some points (such as lines 74-76 are a repetition, plus that concept is never called by its name, namely device fragmentation).

When in line 90 you say "through iterative method and ongoing customer and user feedback" I didn't understand what are you referring to. Probably Agile? Please, clarify.

When you first mentioned OwlEye I didn't understand what you were talking about and had to read the original reference to understand. Please, add a brief summary of what that technique is, how it works and the limitations you bring up. This will allow your paper to be self-contained and the reader to understand your point. A smart thing to do could be to add a related work section with that since I don't think that all of that fits in the introduction. Consider shrinking line 100-106 to the minimum to get to your point, or moving that part to a dedicated section.

line 117 - CNN (convolutional neural network I suppose) was never introduced
line 118 - as far as ourk knowledge extends -- I would say as far as our measurements
line 121 - I don't like superiority, doesn't sound good

The intro of "literature review" section as well doesn't sound very well.
References in lines 135-136 makes the sentence difficult to read, please move them somewhere else.
At some points (i saw at least in line 139, 151, 169) you refer to other studies using "this study", please use "that" instead otherwise it is not clear wether you refer to your own study or what else.

I found confusing you talk about M-UI-R both as a tool and as a concept. Is it an approach or a real tool? Is it available?

In lines 222 and on you talk about RCNN, RPN, ROI without providing any intro and I personally had no idea of what you were talking about, so I think you should provide some context to make the point clearer.

Some references are pretty outdated (Seaman 1999 and Jansen 1998). I suggest the authors try to find some more updated references to support their claim. Other quite old references are those related to the selected baseline: SURF by Bay 2006, SIFT Lowe 2004, Kotsiantis (2007). Here I don't think it is necessary to update the reference, but at least to explain why those approaches are used as a baseline, which is their importance and how do they work. I suggest adding a section in a background section to clarify all those aspects.

Experimental design

Regarding the experimental design, I found that sound, but some choices need to be justified or better explained.

The same applies to the categories of UI smells you selected. Why did you select those four? How did you choose them? Is there a taxonomy from which you choose? Again, expand your discussion about that, because one could argue why you chose that specific set, are those the easier to recognize? It also appears that your finding strictly relate to the category of smell, so with other smells your approach might have different results. Please, also mention that in the threats to validity section (see later)

RQ3: Whether the information ... -> Is the information ...

Among the baselines, four employ ... the remaining two. Didn't you have a total of four baselines? How are 2 remaining? Clarify, please.

"commendable. While the lowest recorded accuracy is 82.5% and the recall value is 77.8%, respectively, the highest precision value is 91.9% and the recall value is 93.8%." is a repetition
"accompanied by a precision of 82.5% and recall of 77.8%, respectively." can be deleted

"This distinction can be attributed to the category’s greater pattern diversity, as well as the relatively smaller size of the buggy region (approximately 8.5% of the component’s area)." This is the most interesting part. Please, expand this providing a more ample discussion.

I didn't understand why line 457 has capitalized words.

Validity of the findings

Lines 426-428: you cannot reason on average if you have that many applications that cannot be explored further than the homepage. You may exclude those from the computation and redo the average, pointing that out.

In the first sentence of the "Data Collection" section, you talk about a dataset collected with a software house. Can you provide more detail about that? Which kind of application, which platform, how was your dataset made and validated? I think this claim needs more details since it is a crucial step to understand the validity and generalizability of the findings.

line 452: A limited-scale experiment was conducted involving three other prominent platforms: can you provide more details about that? Expand if possible please.

A Whole section with threats to validity MUST be inserted. Please, insert that and discuss about those validity threats that may have undermined your research.

---

## Round 0.2 · Minor Revisions

Please revise the document according to reviewer 2's requests.

**Language Note:** The review process has identified that the English language must be improved. PeerJ can provide language editing services - please contact us at [email protected] for pricing (be sure to provide your manuscript number and title). Alternatively, you should make your own arrangements to improve the language quality and provide details in your response letter. – PeerJ Staff

Reviewer 1 ·

Basic reporting

The authors have incorporated my comments/suggestions. I do not have any further comments. The manuscript can be accepted in the current form.

Experimental design

The paper carried out rigorously and results explained very well.

Validity of the findings

Well-supported conclusion and findings based on the experiment.

Reviewer 2 ·

Basic reporting

Thank you for considering prior comments and modifying the paper accordingly. Most of my concerns were fixed, so my suggestions will be related mostly to improve the overall presentation of the paper. First, please check all the open and closed parentheses character, as most of them have spacing issues, being written attached to the previous or next word.

line 51. to enhance the performance -> to enhance the operation. Performances of the applications are not addressed, you are considering mostly UX and faults related.

Line 75. As there are over ten different... -> What do you mean? Probably you should remove "as". This sentence does not make sense as it is.

line 78. You talk about Android fragmentation but you cite a paper dated 2000, when indeed Android probably wasn't even a dream. Revise the citation to stick to what you are referring, or eventually delete this citation.

LIne 84. one the automation.. -> the first is the automation
Line 84. develop -> developed
Line 85 I don't understand why there's a "but". Why have an adversative statement? Where's the problem in the stated approaches?

Line 88. it is very difficult -> However, it is very difficult

Line 109. The first limitation .. too large -> that sentence is difficult to read, subjects are missing. Please rephrase it in a proper and more readable way.
Line 111. that have -> having
Line 112. on al lot -> heavily

Lines 142-146. Those are ok, but harmonize these sentences, otherwise they are not connected and there is no reading flow.

Citations in line from 149-152. This is reprised from the previous review. The problem with these citations is that you are stating too many papers with too many names. Try to split in some way and put the citation at the end of the sentence, otherwise it is challenging for the reader

lines 175-177. the way you illustrate the three RQs is not sound, please try to rephrase them with more harmony.

line 258 - But component occlusion.. -> that's a repetition, maybe say "the latters"

Table 1 is never cited, I also don't think it is necessary, consider removing it or commenting on it properly.

Line 306. We -> we
line 316. The open double quotes are wrong, they are backwards

Line 497. Presented in .. -> not a good start, also is a comparison -> THERE is a comparison
Line 550. Repetition. Pose the question... -> pose them

I don't think a table is useful to represent data in table 5, maybe use a center-aligned stacked bar chart to present these info. In this case, also the questions how effective... do not admit as an answer "agree, disagree ...". Probably the question was Do you think that MUIR is more effective...
Additionally, the percentages are not interesting with just 8 people. Just keep 7 out of 8 and so on.

Line 578. Successfully is repeated.
Line 587-590. Move this limitation to threats to validity.
line 597. conduct -> conducted

Experimental design

Regarding the motivational study I have some comments.

You say that you analyzed screenshots and the corresponding JSON file with static analysis, but at a certain point you mention the XML analysis, I can't understand why? Are they related? Are the XML/JSON a result from a conversion?
Additionally, at a certain point you state that your approach only relies on the images of the GUI to find the flaws and errors in the GUI, but don't you state that you used the JSON? Please, clarify that point.


Line 277. For analysis of text overlap -> for text overlap analysis
Line 280-281. rephrase the last sentence, it is not really sound

In line 282 you state that static analysis has drawbacks, but then why did you make it? You just said that you detected component occlusion in this way, I really don't understand then what you did or not. Clarify this.

RQ3: line 353. You are repeating "is", remove the second
Line 356. As our ... -> As what? the statement has no sense like this, revise it.

Validity of the findings

Validity threats are correctly stated.
The last sentence is really important, and I agree with that.
I would also discuss the different category of application, like testing a streaming application might be different with respect to an accountability/utility/social/chat one. Please, consider adding this.

---

## Round 0.3 · accepted · Accept

Thanks for addressing all the comments. The manuscript is now ready for publication.